

# Intensification of the hydrological cycle expected in West Africa over the 21st century

Stella Todzo[1], Adeline Bichet[2], Arona Diedhiou[1,2]

[1] African Center of Excellence on Climate Change, Biodiversity and Sustainable Agriculture (ACE CCBAD),

University Félix Houphouët Boigny, Abidjan, Côte d'Ivoire.

[2] University Grenoble Alpes, CNRS, IGE UMR 5001, Grenoble, F-38000, France

*Correspondence to*: Adeline Bichet (adeline.bichet@univ-grenoble-alpes.fr)

**Abstract.** This study uses the high resolution outputs of the recent CORDEX-AFRICA climate projections to investigate the future changes in different aspects of the hydrological cycle over West Africa. Over the twenty-first

century, temperatures in West Africa are expected to increase at a faster rate (+ 0.5 °C per decade) than the global average (+ 0.3 °C per decade), and mean precipitation is expected to increase over the Guinea Coast (+ 0.03 mm/day per decade) but decrease over the Sahel (- 0.005 mm/day per decade). In addition, precipitation is expected to become more intense (+ 0.2 mm/day per decade) and less frequent (- 1.5 days per decade) over the entire West Africa as a results of increasing regional temperature (precipitation intensity increases on average by + 0.35 mm/day per °C and

precipitation frequency decreases on average by – 2.2 days per °C). Over the Sahel, the average length of dry spells is also expected to increase with temperature (+ 4% days per °C), which increases the likelihood for droughts with warming in this sub-region. Hence, the hydrological cycle is expected to increase throughout the twenty-first century over the entire West Africa, on average by + 11% per °C over the Sahel as a result of increasing precipitation intensity and lengthening of dry spells, and on average by + 3% per °C over the Guinea Coast as a result of increasing

precipitation intensity only.

## 1 Introduction

It is now established that global warming will result from enhanced anthropogenic greenhouse gases (e.g. Collins et al. 2013). Such a warming is expected to affect precipitation and its variability, especially drought and flood episodes, in

both the tropics and the subtropics (Zwiers et al., 2013; Giorgi et al., 2014). Over West Africa, previous studies (Collins et al., 2013; Diedhiou et al., 2018; Bichet et al., submitted) have shown that the warming is expected to occur at a faster rate than the global average (+ 0.5 vs. + 0.3 °C per decade). Future changes in precipitation however are still unclear (e.g. Collins et al., 2013; Sylla et al., 2016). Future changes in precipitation extremes are expected in some sub-regions, such as an increase in the maximum length of dry spells over West Sahel (Sylla et al., 2016; Diedhiou et al., 2018) and

an intensification of extreme rainfall over the Guinea Coast (Diedhiou et al., 2018). In addition, the growing season is



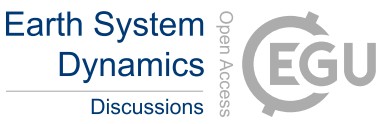

also expected to become shorter in West Africa, while the torrid, arid and semi-arid climate conditions are expected to extend (Sylla et al., 2016). Such conditions can produce significant stresses on agricultural activities, water resources management, ecosystem services and urban areas planning over West Africa, a region that is already highly vulnerable to climate variability. Thus, even though previous studies have reported important changes in the future regional
precipitation, very little is known about the processes involved and the role of the future warming.

The distribution of tropospheric moisture and precipitation is highly complex, but there is one clear and strong control: moisture condensates out of supersaturated air. Assuming that relative humidity would remain roughly constant under global warming, the Clausius-Clapeyron relationship implies that specific humidity would increase exponentially with temperature, at a rate of about 6.5% per °C (e.g. Allen and Ingram, 2002). Assuming no change in the evapo-
transpiration, a warmer atmosphere is thus expected to be able to hold more moisture before reaching saturation, thereby taking more time to reach saturation (longer periods of dryness between two rainy episodes), and releasing more water when moisture does condensate (intensification of the precipitation). Within this integrated view, Giorgi et al. (2011) introduced a single index (HY-INT) that quantitatively combines measures of precipitation intensity and dry spell length, thereby providing an overall metric of hydroclimatic intensity.

To better understand the future impact of the warming on the hydrological cycle in the different sub-regions of West Africa, this study uses the state-of-the-art, high resolution projections of the recent CORDEX-AFRICA experiments to investigate, over the twenty-first century, the future changes in different aspects of the hydrological cycle and their relationship with regional temperatures. After describing the methodology (Section 2), the expected changes in temperature, precipitation, precipitation intensity, dry spells, wet spells, and HY-INT are identified (Section 3.1), before
their relationship with regional temperature is quantified (Section 3.2). Section 4 discusses and concludes the study.

## 2. Dataset and methodology

### 2.1 Methodology

We consider the three following sub-regions: West Sahel (10°N-20°N 18°W-10°W), Central Sahel (10°N-20°N 10°W-10°E), and Guinea Coast (5°N-10°N 10°W-10°E), shown as black boxes in Figure 1a. We focus on annual values over
the period 2006-2099. Following previous studies (Froidurot et al., 2017; Bichet and Diedhiou, 2018a and 2018b), we define a wet (dry) day using the threshold of 1 mm/day. We define a dry spell as a sequence of 2 or more consecutive dry days, that are preceded and followed by a wet day. Hence, the duration of a dry spell, as defined in our study, spans from 2 to 365 days. We compute the annual precipitation intensity (INT), number of wet days (RR1), maximum length of consecutive dry days (CDD), and maximum length of consecutive wet days (CWD) following the definition of the
Expert Team of Climate Change Detection and Indices (ETCCDI; Zhang et al., 2011). Note that because INT



corresponds to the precipitation averaged over wet days, a change in the INT value directly translates into a change in the intensity of wet events, regardless the number of wet events. In addition, we compute the annual contribution of very heavy rain (C98) following Eq. (1):

$$C98 = \frac{PRCPTOT98}{PRCPTOT},$$ (1)

Where $PRCPTOT98$ is the sum of daily precipitation above or equals to the 98[th] percentile annual value at wet days (Pctl98), and $PRCPTOT$ is the sum of daily precipitation at wet days during that year. Following previous studies (Giorgi et al., 2011; Bichet and Diedhiou, 2018a and 2018b), we compute the annual average duration of dry spells (DSL) following Eq. (2):

$$DSL = \frac{NDD}{NDS},$$ (2)

where $NDD$ is the annual number of dry days excluding isolated dry days (single dry day preceded and followed by a wet day), and $NDS$ is the total number of dry spells during that year. Note that the annual number of dry days is directly derived from the annual number of wet days (RR1). Following previous studies (Giorgi et al., 2011; Mohan and Rajeevan, 2017), we compute the annual hydroclimatic intensity index (HY-INT) following Eq. (3):

$$HY-INT = \frac{DSLn}{INTn},$$ (3)

Where $DSLn$ and $INTn$ are the normalized $DSL$ and $INT$, respectively. The normalization consists, for each grid point, in dividing the annual time series of $INT(DSL)$ by its mean value over the period 2006-2099.

**2.2 Data**

We use an ensemble of 18 high-resolution regional climate projections taken from the most up-to-date ensemble produced in the recent years for Africa: CORDEX-AFRICA (Giorgi et al., 2009; Jones et al., 2011). In this ensemble, 5

Regional Climate Models (RCMs) are used to downscale 10 Global Climate Models (GCMs) under the climate scenario RCP8.5 (Table 1). The simulations span the period 2006-2099 at a daily mean time step, and cover Africa (24.64°W – 60.28°E; 45.76°S – 42.24°N) at about 50 km (~ 0.5°) spatial resolution in latitude and longitude. For each simulation and each grid cell, daily time series of surface air temperature and precipitation are retrieved.

We evaluate the spatial distribution of daily rainfall simulated in the CORDEX models by comparing it to the satellite

'Climate Hazards Group Infrared Precipitation with Station data (CHIRPS)' rainfall dataset (Supplementary Figure 1). The CHIRPS dataset is explicitly designed for monitoring agricultural drought and global environmental change over land. It corresponds to a gridded, quasi-global (50° S− 50° N), high-resolution (0.05°), daily rainfall dataset that covers the time period 1981−2014 (Funk et al., 2015). In addition, we evaluate the statistical distribution of daily rainfall simulated in the CORDEX models in three cities (Ouagadougou Airport (12.35° N, 1.52° W), Dakar-Yoff (14.72° N,





17.51° W, and Accra Kotoka International Airport (5.6° N, 0.17° W), as seen in Figure 1a), by comparing it to the CHIRPS dataset (grid point comparison) and to three near-surface daily rain gauge data extracted from the BADOPLUS dataset, as described in Panthou et al. (2012) (Supplementary Figure 2). We find that for all the CORDEX simulations, the spatial distribution of mean precipitation and to some extent RR1 satisfactorily reproduces the spatial distribution of the CHIRPS dataset (Supplementary Figure 1). The multimodel mean however overestimates the observed RR1,

especially along the Guinea Coast. More disagreements are found across models for INT, even though the multimodel mean is in good agreement with the observation. In addition, we find that in Dakar, the CHIRPS and the BADOPLUS datasets show similar statistical distribution of daily mean precipitation, yearly mean precipitation, RR1, and to some extent INT. In Ouagadougou and Accra however, the two observational datasets show similar statistical distributions for daily and yearly mean precipitation, but strongly disagree for RR1 (underestimated in the CHIRPS dataset) and INT

(overestimated in the CHIRPS dataset). In other words, the CHIRPS dataset shows more frequent but less intense precipitation than the BADOPLUS dataset in Ouagadougou and Accra. Finally, we find that the CORDEX simulations generally show a similar statistical distribution than the observations for daily mean and yearly mean precipitation. However, they tend to overestimate (underestimate) the observed RR1 in Accra and Ouagadougou (Dakar), and underestimate the observed (particularly as compared to the BADOPLUS dataset) INT in all three locations. In other

words, the CORDEX simulations show less intense precipitation than the observations (especially the BADOPLUS dataset) in all three locations, and less (more) frequent precipitation in Dakar (Ouagadougou and Accra) than the observations. Nevertheless, we find that the observations are always included within the range of the CORDEX simulations. Hence, we conclude that the CORDEX simulations compare satisfactorily well with the observations, and can be used for the purpose of our study.

**3. Results**

**3.1 Time evolution**

Figure 1 shows the multimodel mean trend maps (2006-2100) for annual a) mean air surface temperature (°C per decade), b) mean precipitation (mm/day per decade), c) INT (mm/day per decade), d) RR1 (days per decade), e) CDD (days), and f) CWD (days) over West Africa. According to Figure 1a, temperature is expected to increase on average by

+ 0.5 °C per decade over the entire region, with a northward increase that reaches + 0.7 °C per decade over the northern Sahel. More specifically, temperature is expected to increase on average by + 0.5, + 0.6, and + 0.45 °C per decade over West Sahel, Central Sahel, and the Guinea Coast, respectively. According to Figure 1b, mean precipitation is expected to increase on average by + 0.03 mm/day per decade over the Guinea Coast, and decrease on average by - 0.015 and - 0.001 mm/day per decade over West Sahel and Central Sahel, respectively. According to Figure 1c, INT is expected to

increase on average by + 0.2 mm/day per decade over the entire West Africa, reaching up to + 0.3 mm/day over the





Guinea Coast. According to Figure 1d, RR1 is expected to decrease on average by - 1.5 days per decade over the entire West Africa, reaching up to - 3 days per decade over West Sahel. According to Figure 1e, CDD is expected to increase on average by + 1 day per decade over the entire region, with a northward increase that reaches + 5 days per decade over the northern Sahel. According to Figure 1f, CWD is expected to decrease on average by – 0.5 day per decade over

the Guinea Coast, and more specifically over Guinea Bissau, Guinea, Sierra Leone, Liberia, Ghana, Benin, Togo and Nigeria. Hence, our results show that by the end of the century, precipitation is expected to intensify but rarefy (including longer dry periods and shorter wet periods) over the entire West Africa, with different impacts on mean precipitation depending on the sub-region: decrease in mean precipitation over the Sahel (especially over West Sahel) and increase in mean precipitation over the Guinea Coast.

Figure 2 shows the multimodel mean trend maps (2006-2100) for annual a) INTn, b) DSLn, and c) HY-INT. In agreement with Figure 1c, Figure 2a shows an intensification of precipitation over the entire West Africa, on average by + 0.02 (2 %) per decade. According to Figure 2b, DSL is expected to increase over the Sahel on average by + 0.05 (5 %) per decade, with a latitudinal increase northward that reaches + 0.1 (10 %) per decade over the western part of the Sahel. Negligible changes are expected over the Guinea Coast. According to Figure 2c, HY-INT is expected to increase

on average by + 0.05 (5 %) over West Africa, with a latitudinal increase northward that reaches + 1.5 (15 %) per decade over the Sahel. Hence, our results show that the hydrological cycle is expected to intensify by the end of the century over the entire West Africa. Over the Guinea Coast, this intensification results exclusively from an increase in precipitation intensity (INT). Over the Sahel however, it results from both, an increase in precipitation intensity (INT) and an increase in the average length of dry spells (DSL). Hence over the Sahel, rainfall events are expected to become

more intense and separated by much longer periods of dryness.

### 3.2 Relationship with temperature

Figure 3 shows the annual values, over 2006-2099, of mean precipitation (mm/day, top row), RR1 (days, middle row), and INT (mm/day, bottom row), plotted against the corresponding annual mean values of air surface temperature (°C), and as averaged over a) West Sahel, b) Central Sahel and c) Guinea Coast. Individual CORDEX simulations are

represented by the thin colored dots (see Table 1 for color references), and the multimodel mean is represented by the thick black dots. According to Figure 3, mean precipitation decreases with temperature over the Sahel (multimodel mean decreases by - 0.032 and - 0.012 mm/day per °C over West Sahel and Central Sahel, respectively) and increases with temperature over the Guinea Coast (multimodel mean increases by + 0.062 mm/day per °C). In all three sub-regions, RR1 decreases with temperature (multimodel mean decreases by - 2.3, - 1.7 and - 2.6  days per °C over West

Sahel, Central Sahel and Guinea Coast, respectively) while INT increases with temperature (multimodel mean increases by + 0.33, + 0.27 and + 0.41 mm/day per °C, respectively). Based on the multimodel mean values, we find that changes





in temperature explain less than 24 % of the changes in mean precipitation in all three sub-regions, but more than 67 % (51 %) of the changes in RR1 (INT). According to Figure 3, even though the annual mean values vary greatly from a simulation to another (e.g. NCC-NorESM1-HIRHAM5 is particularly warm over Central Sahel, ICHEC-RACMO is

particularly cold over the three sub-regions, NCC-NorESM1-HIRHAM5 is particularly wet over the Guinea Coast, and CSIRO-SMHI is particularly wet over the Sahel), the relation between each variable and the temperature is consistent across all models, albeit with a different strength.

Figure 4 shows the annual values, over 2006-2099, of INTn (top row), DSLn (middle row), and HY-INT (bottom row), plotted against the corresponding annual mean values of air surface temperature (°C), and as averaged over a) West

Sahel, b) Central Sahel, and c) Guinea Coast. As for Figure 3, individual CORDEX simulations are represented by the thin colored dots and the multimodel mean by the thick black dots. According to Figure 4, INTn increases with temperature in all three regions (multimodel mean increases by + 0.031 (3.1 %) per °C over the Sahel and + 0.041 (4.1 %) per °C over the Guinea Coast). However, whereas DSLn increases with temperature over the Sahel (multimodel mean increases by + 0.093 (9.3 %) and + 0.056 (5.6 %) per °C over West Sahel, and Central Sahel, respectively), it

decreases with temperature over the Guinea Coast (multimodel mean decreases by – 0.01 (1 %) per °C). Finally, HY-INT increases with temperature in all three sub-regions (multimodel mean increases by + 0.13 (13 %), + 0.089 (8.9 %), and + 0.031 (3.1 %) per °C over West Sahel, Central Sahel, and the Guinea Coast, respectively). Based on the multimodel mean values, we find that changes in temperature explain 63 % (74 %) of the changes in DSLn (HY-INT) over the Sahel and 14 % (27 %) over the Guinea Coast. According to Figure 4, the annual mean values of DSLn vary

greatly from a simulation to another over the Sahel, but are similar across simulations over the Guinea Coast. As for Figure 3, we find that the relation between each variable and the temperature is consistent across all models, albeit with different strength. We conclude that most of the trends observed in Figures 1 and 2 show a positive relationship with regional warming.

### 4 Discussion and conclusion

This study uses an ensemble of high resolution regional climate projections (CORDEX-AFRICA) to investigate, over the twenty-first century, the relationship between regional warming and different aspects of the hydrological cycle, as seen in three different sub-regions of West Africa. In agreement with previous studies (e.g. Vizy and Cook, 2012; Collins et al., 2013; Sylla et al., 2016; Diedhiou et al., 2018; Klutse et al., 2018), we find that 1) West African surface temperatures are expected to increase at a faster rate than the global averaged warming (+ 0.5 °C vs. +

0.3 °C per decade), 2) precipitation is expected to intensify but rarefy over the entire region, 3) dry spells are expected to become longer (especially over the northern and the western part of the Sahel), and 4) wet spells are expected to become shorter over the Guinea Coast.





In addition, we show that 1) mean precipitation is expected to increase over the Guinea Coast and decrease over the Sahel, and 2) the hydrological cycle, as defined by Giorgi et al. (2011), is expected to intensify over the entire

West Africa (+ 5 % per decade on average). Whereas this intensification results solely from more intense precipitation over the Guinea Coast, we find it results from both, more intense precipitation (+ 2 %) and longer periods of dryness (+ 5-10 %) over the Sahel.

According to our results, all the aforementioned trends show a positive relationship with regional temperatures. In agreement with Collins et al. (2013), we find that mean precipitation is expected to decrease with temperature over

the Sahel and increase with temperature over the Guinea Coast. In addition, we find that the hydrological cycle is expected to increase with temperature over the entire West Africa, on average by + 11 % per °C over the Sahel and + 3 % per °C over the Guinea Coast. This increase is in qualitative agreement with the Clausius-Clapeyron relationship, which implies that specific humidity would increase exponentially with temperature (at a rate of about 6.5% per °C), meaning that a warmer atmosphere is expected to take longer to reach saturation and release more water when it

condensates, thereby intensifying the hydrological cycle (e.g. Allen and Ingram, 2002). Over the Sahel, we find indeed that the warmer atmosphere takes longer to reach saturation (DSL increases on average by + 7.5 % per °C) and releases more water when it condensates (INT increases on average by + 3.1 % per °C). Over the Guinea Coast however, we find that the warmer atmosphere does not take longer to reach saturation (DSL decreases on average by − 1 % per °C), but does release more water when it condensates (INT increases by + 4.1 % per °C but DSL decreases by − 1 % per °C).

To understand the processes involved, Figure 5 (top row) shows the evolution of annual mean specific humidity as a function of annual mean temperatures in all three sub-regions. According to Figure 5, the specific humidity increases with temperature on average by + 5 % per °C in all three sub-regions, which is close to the rate expected from the Clausius-Clapeyron relationship (+ 6.5 % per °C). Thus, we conclude that in all three sub-regions, a warmer atmosphere does increase the amount of moisture in the atmosphere, which leads to more intense precipitation (INT increase in both

sub-regions). However, whereas a warmer atmosphere also leads to longer periods of dryness over the Sahel (DSL increase over the Sahel), this is not the case over the Guinea Coast. We suggest that unlike the Sahel, the atmosphere over the Guinea Coast does not require more time reach saturation because it is already very close to saturation. Thus, although the likelihood for droughts increases with temperature over the Sahel (and in particular over West Sahel), this is not the case over the Guinea Coast. To understand the impact on the very heavy rainfall and floods, Figure 5 also

shows the evolution of the 98th percentile annual value (Pctl98 in mm/day, middle row) and the annual contribution of very heavy rain (C98 in %, bottom row) as a function of annual mean temperatures in all three sub-regions. According to Figure 5, a warmer climate implies heavier rainfall (multimodel mean increases by + 3.1 %, + 4.2 %, and + 8.6 % over West Sahel, Central Sahel, and Guinea Coast, respectively) and a larger contribution of very heavy rainfall (+ 5.6



%, + 4.1 %, and + 3.7 % in West Sahel, Central Sahel, and Guinea Coast, respectively) in all three sub-regions, which

indicates an increase in the likelihood for floods with temperature over the entire West Africa.

Finally, it is worth noting that over the Sahel, precipitation intensity is also driven by the frequency of the Mesoscale Convective Systems (MCSs), which are driven by the meridional temperature gradient between the Sahel and the Sahara (Taylor et al., 2017). Whereas the meridional temperature gradient between the Sahel and the Sahara is projected to increase in CORDEX-AFRICA (on average by + 2 °C by the end of the century, not shown), the impact of

this increase on the frequency of the MCSs cannot be simulated by these models (50 km) because MCSs occur on scales that are not resolved by these models. Hence, we suggest that over the Sahel, precipitation intensity may increase more than projected in our study, as result of the increasing meridional temperature gradient between the Sahel and the Sahara. Additional simulations at higher resolution would be required to confirm this hypothesis.

**Figure Captions.**

**Table 1.** Summary of 18 simulations (GCM/RCM chains) taken from the CORDEX-AFRICA data. In this ensemble, 5 RCMs are used to downscale 10 GCMs. Each experiment comprises one historical and one scenario (RCP8.5) run, spanning the periods 1981-2005 and 2006-2099 respectively. The horizontal resolution of all simulations is 0.5° in both latitude and longitude. The colors and symbols refer to Figures 3, 4, and 5.

**Figure 1.** Multimodel mean trend maps (2006-2100) for annual a) mean temperature (°C per decade), b) mean precipitation (mm/day per decade), c) INT (mm/day per decade), d) RR1 (days per decade), e) CDD (days per decade), and f) CWD (days per decade). Trends that are not significant at 95% according to the Student's t-test are shaded in gray. The black boxes correspond to the three regions of interested: West Sahel, (10°N-20°N 18°W-10°W) Central Sahel (10°N-20°N 10°W-10°E), and the Guinea Coast (5°N-10°N 10°W-10°E), respectively.

**Figure 2.** Multimodel mean trend maps (2006-2100) for annual a) INTn, b) DSLn, and c) HY-INT, in unit per decade. Trends that are not significant at 95% according to the Student's t-test are shaded in gray.

**Figure 3.** Annual values of mean precipitation (mm/day), RR1 (days), and INT (mm/day) (y-axis) shown against annual mean temperature (°C) (x-axis), and averaged over a) West Sahel, b) Central Sahel, and c) Guinea Coast. Each color corresponds to a single simulation, as described in Table 1, and the thick black dots correspond to the multimodel mean.

Also shown are the fitted regression line of the multimodel mean (red line) and the associated coefficient of determination ('$r^2$') and correlation ('slope').

**Figure 4.** Annual values of INTn, DSLn, and HY-INT (y-axis) shown against annual mean temperature (°C) (x-axis), and averaged over a) West Sahel, b) Central Sahel, and c) Guinea Coast. Each color corresponds to a single simulation,

as described in Table 1, and the thick black dots correspond to the multimodel mean. Also shown are the fitted regression line of the multimodel mean (red line) and the associated coefficient of determination ('$r^2$') and correlation ('slope').

**Figure 5.** Annual values of specific humidity, 98[th] percentile (mm/day), and contribution of precipitation above the 98[th] percentile (%) (y-axis)shown against annual mean temperature (°C) (x-axis), and averaged over a) West Sahel, b) Central Sahel, and c) Guinea Coast. Each color corresponds to a single simulation, as described in Table 1, and the thick 250 black dots correspond to the multimodel mean. Also shown are the fitted regression line of the multimodel mean (red line) and the associated coefficient of determination ('$r^2$') and correlation ('slope', in % per °C, as compared to the 2006-2100 mean value). Note that specific humidity for the model NCC-NorESM1-HIRHAM5 was not available to download at the time of the analysis.

**Data availability:** The CORDEX-AFRICA data set from the World Climate Research Program's Working Group on Regional Climate is freely available to download at http://www.cordex.org/data-access/esgf/). The CHIRPS dataset from the Climate Hazards Group is freely available to download at http://chg.geog.ucsb.edu/data/chirps. The BADOPLUS dataset is freely available to download at http://www.amma-catch.org/.

**Author contribution:** The authors declare to have no conflict of interest with this work. A. Bichet and A. Diedhiou fixed the analysis framework. S. Todzo carried out all the calculations and analyses and produced graphs. All authors contributed to the redaction.

**Acknowledgements:** The research leading to this publication received funding from the African Center of Excellence 265 on Climate Change, Biodiversity and Sustainable Agriculture (ACE CCBAD), the NERC/DFID "Future Climate for Africa" program under the AMMA-2050 project, grant number NE/M019969/1, and the French public research institution IRD (Institut de Recherche pour le Développement; France). This work is a contribution to CORDEX-AFRICA initiative. We acknowledge the usage of the CORDEX-AFRICA data set from the World Climate Research Program's Working Group on Regional Climate (http://www.cordex.org/data-access/esgf/) and the CHIRPS dataset 270 from the Climate Hazards Group (http://chg.geog.ucsb.edu/data/chirps). Our work has benefited from access to rainfall data sets provided by the AMMA-CATCH observatory, the AMMA international program, DMN Burkina, ANACIM, and DMN Niger; we sincerely thank all of them, as well as the staff at the IGE computation center (Guillaume Quantin, Véronique Chaffard, Patrick Juen, and Wajdi Nechba) for their technical support,



and Geremy Panthou for his role in accessing the data and insights into the dataset. We also thank the staff at the

IGE computation center (Patrick Juen and Wajdi Nechba) for their technical support.

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





**Table 1.** Summary of 18 simulations (GCM/RCM chains) taken from the CORDEX-AFRICA data. In this ensemble, 5
RCMs are used to downscale 10 GCMs. Each experiment comprises one historical and one scenario (RCP8.5) run,
spanning the periods 1981-2005 and 2006-2099 respectively. The horizontal resolution of all simulations is 0.5° in both
latitude and longitude. The colors and symbols refer to Figures 3, 4, and 5.

| RCM \ GCM | DMI-HIRHAM5_v2 | CLMcom-CCLM4-8-17_v1 | KNMI-RACMO22T_v1 | SMHI-RCA4_v1 | REMO2009_v1 |
|---|---|---|---|---|---|
| ICHEC-EC-EARTH | (.) | (.) | (+) | (.) | (.) |
| CNRM-CERFACS-CNRM-CM5 | | (.) | | | |
| MPI-M-MPI-ESM-LR | | (+) | | (+) | (+) |
| NCC-NorESM1-M | (+) | | | (.) | |
| NOAA-GFDL-GFDL-ESM2M | | | | (.) | |
| IPSL-IPSL-CM5A-MR | | | | (+) | (+) |
| MIROC-MIROC5 | | | | (+) | |
| CSIRO-QCCCE-CSIRO-Mk3-6-0 | | | | (.) | |
| CCCma-CanESM2 | | | | (.) | |
| MOHC-HadGEM2-ES | | (+) | | | |

**Figures.**

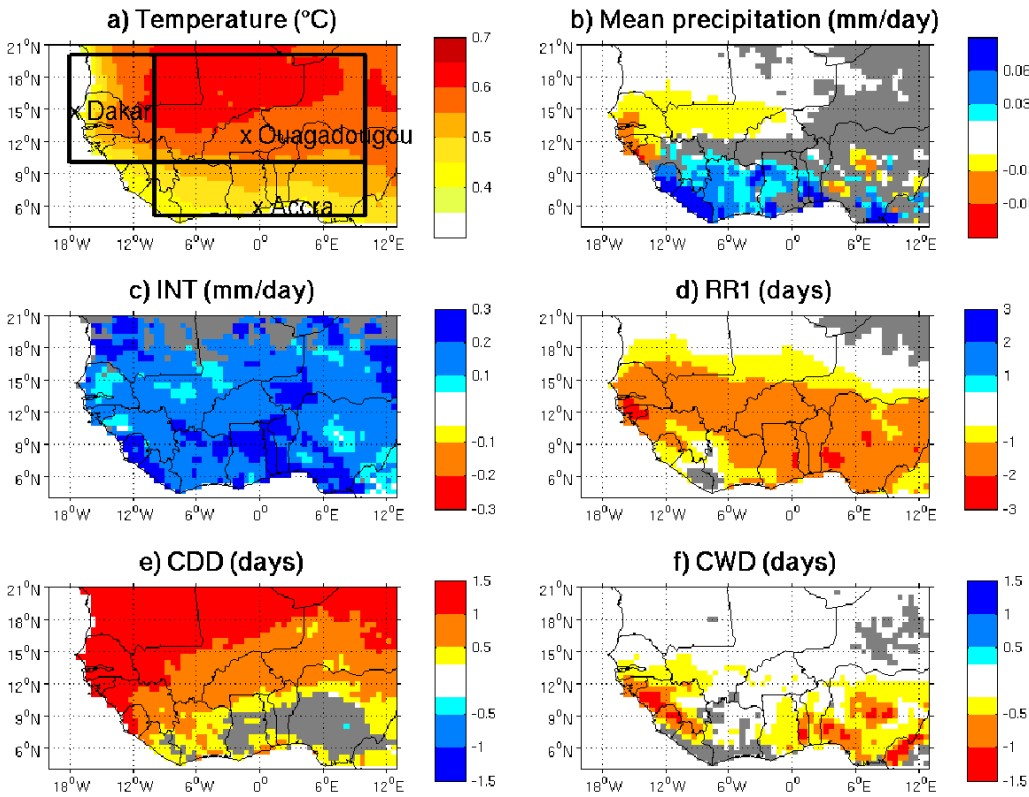

**Figure 1.** Multimodel mean trend maps (2006-2100) for annual a) mean temperature (°C per decade), b) mean precipitation (mm/day per decade), c) INT (mm/day per decade), d) RR1 (days per decade), e) CDD (days per decade), and f) CWD (days per decade). Trends that are not significant at 95% according to the Student's t-test are shaded in gray. The black boxes correspond to the three regions of interested: West Sahel, (10°N-20°N 18°W-10°W) Central Sahel (10°N-20°N 10°W-10°E), and the Guinea Coast (5°N-10°N 10°W-10°E), respectively.





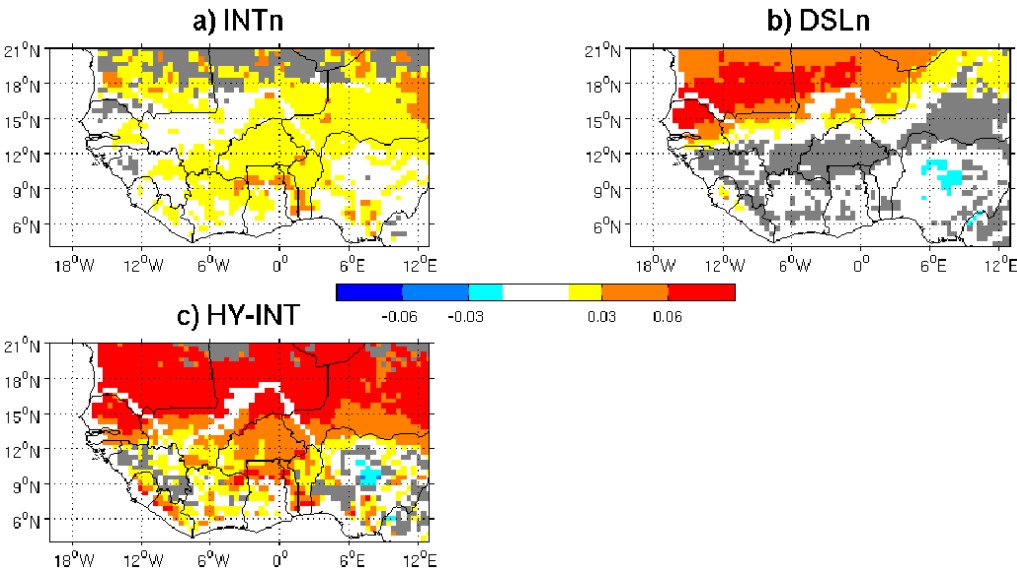

**Figure 2.** Multimodel mean trend maps (2006-2100) for annual a) INTn, b) DSLn, and c) HY-INT, in unit per decade.

Trends that are not significant at 95% according to the Student's t-test are shaded in gray.







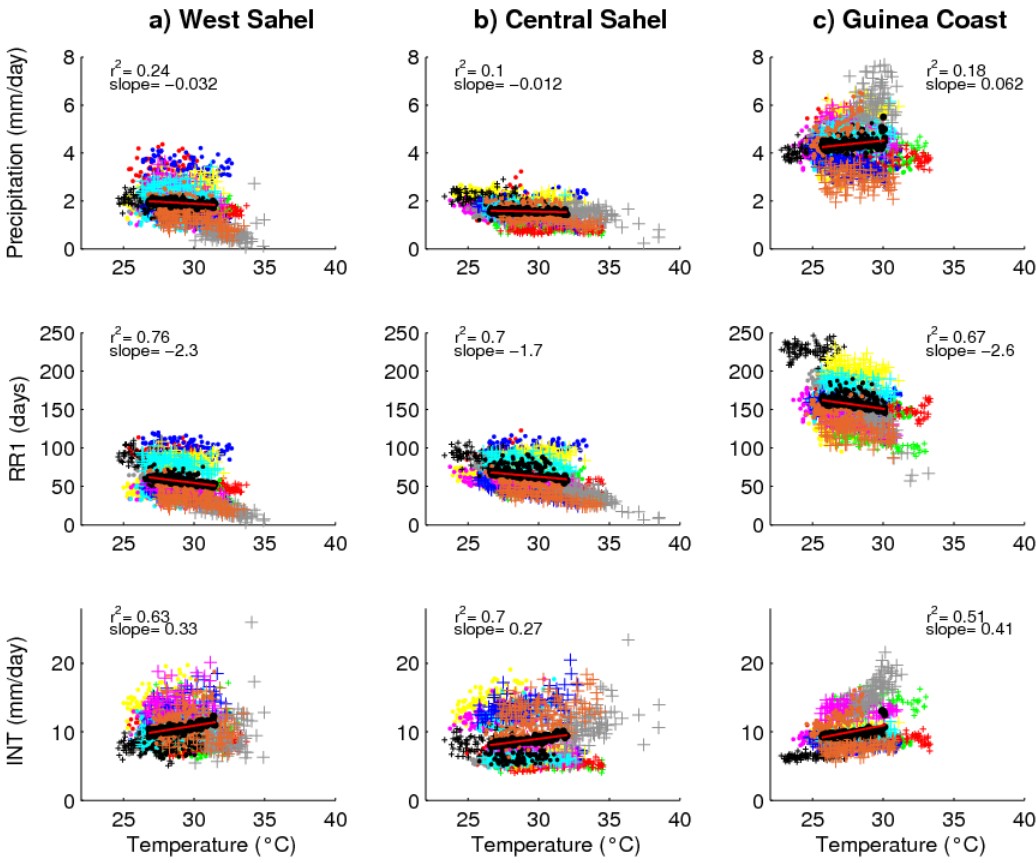

**Figure 3.** Annual values of mean precipitation (mm/day), RR1 (days), and INT (mm/day) (y-axis) shown against annual mean temperature (°C) (x-axis), and averaged over a) West Sahel, b) Central Sahel, and c) Guinea Coast. Each color corresponds to a single simulation, as described in Table 1, and the thick black dots correspond to the multimodel mean. Also shown are the fitted regression line of the multimodel mean (red line) and the associated coefficient of determination ('$r^2$') and correlation ('slope').






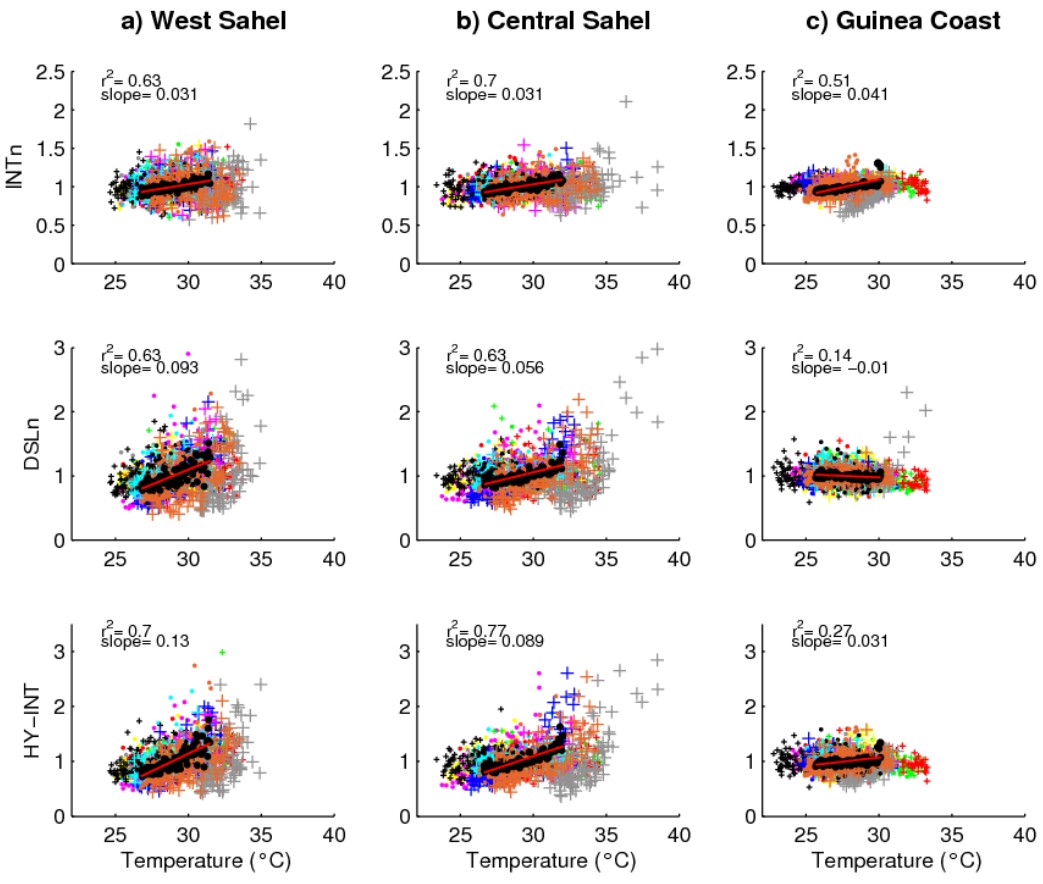

**Figure 4.** Annual values of INTn, DSLn, and HY-INT (y-axis) shown against annual mean temperature (°C) (x-axis), and averaged over a) West Sahel, b) Central Sahel, and c) Guinea Coast. Each color corresponds to a single simulation, as described in Table 1, and the thick black dots correspond to the multimodel mean. Also shown are the fitted regression line of the multimodel mean (red line) and the associated coefficient of determination ('$r^2$') and correlation ('slope').

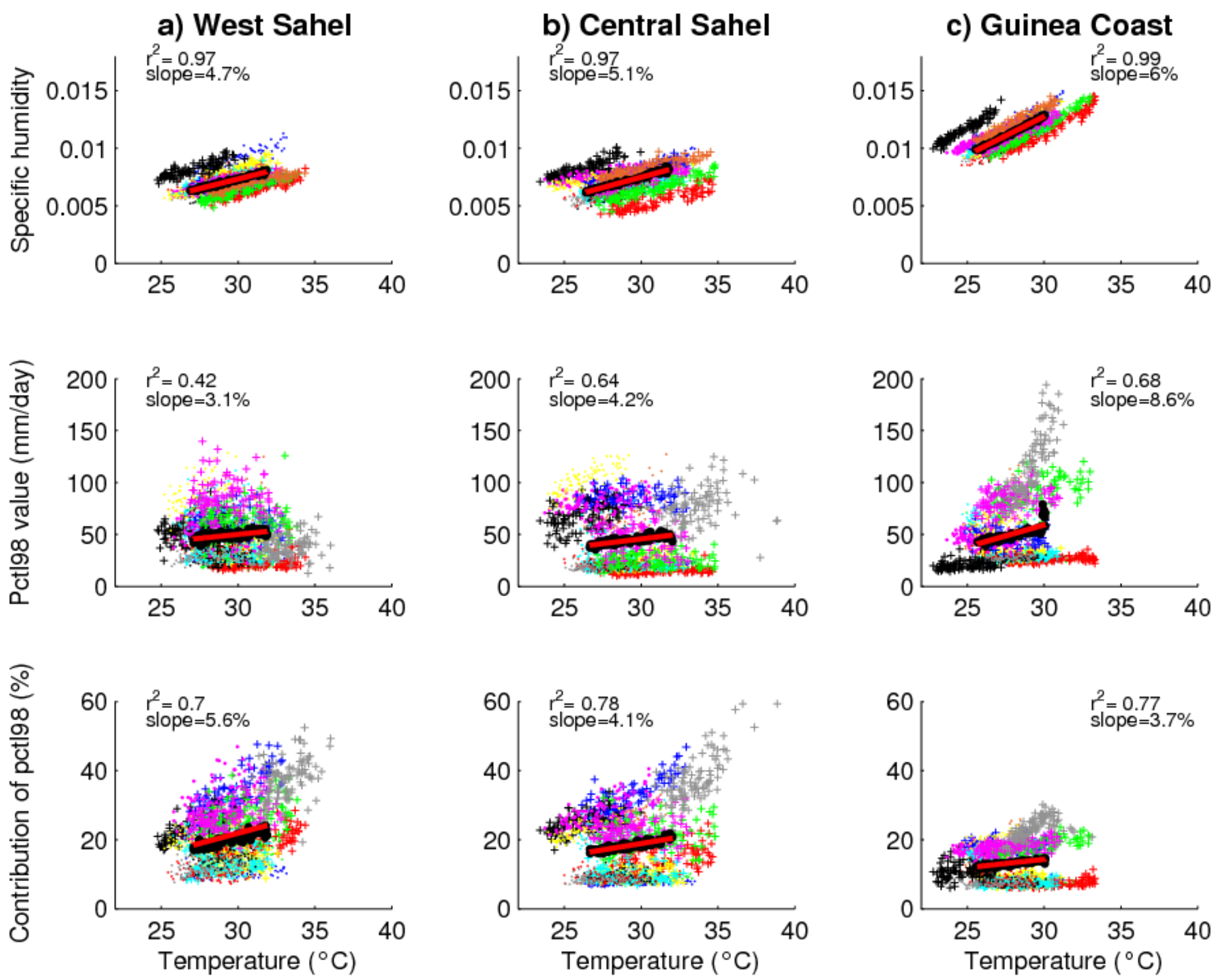

**Figure 5.** Annual values of specific humidity, 98th percentile (mm/day), and contribution of precipitation above the 98th percentile (%) (y-axis)shown against annual mean temperature (°C) (x-axis), and averaged over a) West Sahel, b) Central Sahel, and c) Guinea Coast. Each color corresponds to a single simulation, as described in Table 1, and the thick black dots correspond to the multimodel mean. Also shown are the fitted regression line of the multimodel mean (red line) and the associated coefficient of determination ('$r^2$ ') and correlation ('slope', in % per °C, as compared to the 2006-2100 mean value). Note that specific humidity for the model NCC-NorESM1-HIRHAM5 was not available to download at the time of the analysis.