# Peer review of "Intensification of the hydrological cycle expected in West Africa over the 21st century"

_Earth System Dynamics, 2019_

## Referee Comment (RC1) · Anonymous Referee #1 · 9 Nov 2019

Reviewer comment on Intensification of the hydrological cycle expected in West Africa over the 21st century by Todzo et al. General comment This study investigated the hydrological cycle over West African region using CORDEX-AFRICA data between 2006 and 2099. To achieve their purpose, authors used the known ETCCD indexes combined with hydroclimatic indice. They found that West African region is expected to be warmer than the global average per decade (0.5o C vs 0.3o C ). Also precipitations are expected to become more intense and less frequent. This is a very interesting work and address relevant scientific questions in the scope of ESD. Data and methodology are clearly stated as well as main findings. However the writing style is too amorphous from beginning to the end making this interesting fining a bit boring on reading. Please below see my suggestions to improve the manuscript:

[Figure]

Major comment Authors discussed how the MCSs may impact precipitation distribution over the studied though these MCSs were not considered in the CORDEX data. A similar attempt should be done for others forcing such as aerosols especially dust that are abundantly present in the region and known to impact West African climate through their radiative forcing: Konare, A., et al. "A regional climate modeling study of the effect of desert dust on the West African monsoon." Journal of Geophysical Research: Atmospheres 113.D12 (2008). N'Datchoh, E. T., et al. "Dust induced changes on the West African summer monsoon features." International Journal of Climatology 38.1 (2018): 452-466. Yoshioka, Masaru, et al. "Impact of desert dust radiative forcing on Sahel precipitation: Relative importance of dust compared to sea surface temperature variations, vegetation changes, and greenhouse gas warming." Journal of Climate 20.8 (2007): 1445-1467. Silué et al., (2019) Evidence of Long-Term Trend of Visibility in the Sahel and Coevolution with Meteorological Conditions and Vegetation Cover during the Recent Period. Atmospheric and Climate Sciences, 9, 346-368. doi: 10.4236/acs.2019.93025.

Minor comments L30 - L32 "Growing season": I guess it refers to agriculture however, as stated links are that clear for the reader. I suggest authors to include a sentence relating precipitation to the agriculture before this statement. L36: Authors stated "tropospheric moisture and precipitation is highly complex" Is it in the world or very specific to the studied region? Please specify it. L72: please change "Following" by or "base on" it looks quite monotone to use "following" L92: Please add a connector between statements to make the test flow L102: please replace "than" by "with" L112 – L114: please edit the sentence by making it more clear. The statement is confusing and look more like figure label. Across the entire manuscript, authors stated "according to" more than 10 times when commenting and discussing figures. It makes the manuscript quite monotone I suggest some of these should be change may be rewrite a little bit differently using statement such as: From Figure . . . Figure . . . suggests /implies . . . (Not exhaustive)

[Figure]

Though the active form writing style is correct and show the authors efforts and work, it may be interesting to have some variation in the manuscript with the passive form too.

Similarly, all figures comments start by "Figure xx shows" however these statements can be write differently to allow the manuscript to be more alive. For example, L158-L160: Statement can be rewrite as: "The annual values . . .. are shown in figure 4"

L162: All three regions please replace by "all the three regions" L169: please write "explain about" L170: please replace "as for" by "similarly to"

---

## Referee Comment (RC2) · Anonymous Referee #2 · 22 Jan 2020

Review of "intensification of the hydrological cycle expected in West Africa over the 21st century" by S. Todzo, A. Bichet & A Diedhiou

This paper looks at the intensification of the hydrological cycle over West Africa, a small part of the CORDEX Africa region, using a number of the CORDEX simulations. Results are given for the multi-model mean.

The paper is clearly written and describes the observational and model datasets used. The methods used to analysed the data are not novel but are clearly described and referenced. The supplementary data is used to show how well the individual regional simulations and the multi-model mean for the current climate agree with observations.

One thing the paper omits is any justification of using a multi-model mean and any

discussion of the set of CORDEX regional simulations chosen in Table 1. 8 of the 18 members using the same regional model SMHI-RCA4_v1. Does this impact on the results? Looking at Figure 3 to 5 it looks as though some of the results from the other regional model are on the edges of the ensemble of points.

Dosio et al 2019 (Climate Dynamics) is not referenced and gives a comparison of the CORDEX Africa regional model results.

Knutti et al 2010 (J Clim) Challenges in combining Projections from multiple models.

Specific Comments:

Page 2, L46 The first mention of CORDEX-AFRICA has no references.

Page 8 L 221 There are now published results for West Africa from a convection permitting resolution simulation over Africa. Berthou et al 2019 (Geophy Res Let).

---

## Author Response (AR1)

**Response to Reviewer 1**

We thank the reviewer for the constructive comments. We appreciate the time and effort that the reviewer has dedicated to providing valuable feedbacks on our manuscript. All the comments have been accounted for.

**Comments from Reviewer 1**

*This study investigated the hydrological cycle over West African region using CORDEX-AFRICA data between 2006 and 2099. To achieve their purpose, authors used the known ETCCD indexes combined with hydroclimatic indice. They found that West African region is expected to be warmer than the global average per decade (0.5o C vs 0.3o C ). Also precipitations are expected to become more intense and less frequent. This is a very interesting work and address relevant scientific questions in the scope of ESD. Data and methodology are clearly stated as well as main findings. However the writing style is too amorphous from beginning to the end making this interesting fining a bit boring on reading. Please below see my suggestions to improve the manuscript.*

**Major comment**
**Comment 1**: *Authors discussed how the MCSs may impact precipitation distribution over the studied though these MCSs were not considered in the CORDEX data. A similar attempt should be done for others forcing such as aerosols especially dust that are abundantly present in the region and known to impact West African climate through their radiative forcing: Konare, A., et al. "A regional climate modeling study of the effect of desert dust on the West African monsoon." Journal of Geophysical Research:Atmospheres 113.D12 (2008). N'Datchoh, E. T., et al. "Dust induced changes on the West African summer monsoon features." International Journal of Climatology 38.1 (2018): 452-466. Yoshioka, Masaru, et al. "Impact of desert dust radiative forcing on Sahel precipitation: Relative importance of dust compared to sea surface temperature variations, vegetation changes, and greenhouse gas warming." Journal of Climate 20.8 (2007): 1445-1467. Silué et al., (2019) Evidence of Long-Term Trend of Visibility in the Sahel and Coevolution with Meteorological Conditions and Vegetation Cover during the Recent Period. Atmospheric and Climate Sciences, 9, 346-368. doi:10.4236/acs.2019.93025.*
**Response:** We agree with the reviewer, and such a discussion is now included in the revised manuscript as follows: "Similarly, the impacts of atmospheric aerosols, particularly abundant over West Africa due to seasonal desert dusts (Konare et al. 2008, N'Datchoh et al. 2018), are only partially accounted for in CORDEX AFRICA due to the simplified parameterization schemes for aerosols in this dataset. However, because aerosols are expected to affect temperature and precipitation in this region (e.g. Konare et al. 2008, N'Datchoh et al. 2018), we suggest that our results are also limited by this simplified representation of aerosols. Additional simulations at higher resolution and using a more complex parameterization scheme for aerosols would be required to identify the impact of MCSs and aerosols on our results, which is beyond the scope of our study."

**Minor comments**

**Comment 2:** *L30 - L32 "Growing season": I guess it refers to agriculture however, as stated links are that clear for the reader. I suggest authors to include a sentence relating precipitation to the agriculture before this statement.*
**Response**: Thank you for pointing this out. This is now clarified in the revised manuscript as follows: "Particularly relevant for agriculture, changes in precipitation are also projected during the growing season, expected to become shorter, as torrid, arid, and semi-arid climate conditions are expected to extend (Sylla et al., 2016)."

**Comment 3:** L36: *Authors stated "tropospheric moisture and precipitation is highly complex" Is it in the world or very specific to the studied region? Please specify it.*
**Response**: Global, this is now clarified in the revised manuscript.

**Comment 4:** *L72: please change "Following" by or "base on" it looks quite monotone to use "following"*
**Response**: The revised manuscript has been altered accordingly.

**Comment 5**: *L92: Please add a connector between statements to make the test flow*
**Response**: Thank you for this point, the following sentenced has been added in the revised manuscript: "Results are shown in Supplementary Figures 1 and 2, and summarized here. Each for the 18 CORDEX simulations satisfactorily reproduces the spatial distribution of the CHIRPS daily mean precipitation...".

**Comment 6:** *L102: please replace "than" by "with"*
**Response:** The revised manuscript has been altered accordingly.

**Comment 7**: *L112 – L114: please edit the sentence by making it more clear. The statement is confusing and look more like figure label.*
**Response**: We agree with the reviewer, the revised manuscript has been altered as follows: "Trends (2006-2100) of air surface temperature (°C per decade), mean precipitation (mm/day per decade), INT (mm/day per decade), RR1 (days per decade), CDD (days), and CWD (days) are shown in Figure 1, as multimodel mean maps. From Figure 1, temperature…".

**Comment 8**: *Across the entire manuscript, authors stated "according to" more than 10 times when commenting and discussing figures. It makes the manuscript quite monotone I suggest some of these should be change may be rewrite a little bit differently using statement such as: From Figure : : : Figure : : : suggests /implies : : : (Not exhaustive)*
**Response**: We agree with the reviewer, the revised manuscript has been modified throughout accordingly. Here are a few examples:
E.g. 1: "Mean precipitation is expected to increase on average by + 0.03 mm/day per decade over the Guinea Coast, and decrease on average by - 0.015 and - 0.001 mm/day per decade over West Sahel and Central Sahel, respectively (Figure 1b). INT is expected…".
E.g. 2: "Trends (2006-2100) of INTn, DSLn, and HY-INT are shown in Figure 2, as multimodel mean maps.".
E.g. 3: "In addition, whereas DSL is expected to increase over the Sahel on average by + 0.05 (5 %) per decade with a latitudinal increase northward that reaches + 0.1 (10 %) per decade over the western part of the Sahel, negligible changes are expected over the Guinea Coast (Figure 2b). As a result, HY-INT is…".
E.g. 4: "Annual values (2006-2009) of mean precipitation (mm/day), RR1 (days), and INT (mm/day) are shown in Figure 3, plotted against the corresponding annual mean values of air surface temperature (°C), and as averaged over a) West Sahel, b) Central Sahel, and c) Guinea Coast."
E.g. 5: "As seen in Figure 3, even though…".

**Comment 9**: *Though the active form writing style is correct and show the authors efforts and work, it may be interesting to have some variation in the manuscript with the passive form too. Similarly, all figures comments start by "Figure xx shows" however these statements can be write differently to allow the manuscript to be more alive. For example, L158-L160: Statement can be rewrite as: "The annual values : : :. are shown in figure 4"*

**Response**: We agree with the reviewer, the revised manuscript has been modified accordingly throughout. For example, the sentence you noted now reads as follows: "Annual values (2006-2009) of INTn, DSLn, and HY-INT are shown in Figure 4, plotted against..."

**Comment 10**: *L162: All three regions please replace by "all the three regions"*
**Response:** The revised manuscript has been altered accordingly.

**Comment 11**: *L169: please write "explain about"*
**Response:** The revised manuscript has been altered accordingly.

**Comment 12**: *L170: please replace "as for" by "similarly to"*
**Response:** The revised manuscript has been altered accordingly.
We thank the reviewer for the constructive comments. We appreciate the time and effort that the reviewer has dedicated to providing valuable feedbacks on our manuscript. All the comments have been accounted for.

**Response to Reviewer 2**

*This paper looks at the intensification of the hydrological cycle over West Africa, a small part of the CORDEX Africa region, using a number of the CORDEX simulations. Results are given for the multi-model mean. The paper is clearly written and describes the observational and model datasets used. The methods used to analyzed the data are not novel but are clearly described and referenced. The supplementary data is used to show how well the individual regional simulations and the multi-model mean for the current climate agree with observations.*

**Comment 1**: *One thing the paper omits is any justification of using a multi-model mean and any discussion of the set of CORDEX regional simulations chosen in Table 1. 8 of the 18 members using the same regional model SMHI-RCA4_v1. Does this impact on the results? Looking at Figure 3 to 5 it looks as though some of the results from the other regional model are on the edges of the ensemble of points.*
*Dosio et al 2019 (Climate Dynamics) is not referenced and gives a comparison of the CORDEX Africa regional model results.*
*Knutti et al 2010 (J Clim) Challenges in combining Projections from multiple models.*

**Response**: We thank the reviewer for this constructive comment and useful references. They have all been accounted for in the revised manuscript as follows: "We use an ensemble of 18 high-resolution regional climate projections taken from the most up-to-date ensemble produced in the recent years for Africa: CORDEX-AFRICA (Giorgi et al., 2009; Jones et al., 2011; Hewitson et al. 2012; Kim et al. 2014). All the simulations available online at the time of the analysis have been used. In this ensemble, 5 Regional Climate Models (RCMs) are used to downscale 10 Global Climate Models (GCMs) under the climate scenario RCP8.5 (Table 1). Out of the 50 combinations possible, only 18 were available, from which 8 use the same RCM. Whereas this imbalance presents the disadvantage to slightly bias the results towards this RCM, it also presents the advantage of representing a large number of GCM, not accessible otherwise. Because the impact of the heterogeneity of the CORDEX-AFRICA GCM-RCM matrix on future precipitation changes is found mostly over Central and West Africa (Dosio et al., 2019), we choose to represent a maximum diversity of RCMs and GCMs. Furthermore, although averaging model output may lead to a loss of signal (such that the true expected change is very likely to be larger than suggested by a model average), there is too little agreement on metrics to separate "good" and "bad" models to objectively weight the models (Knutti et al., 2010). In the following, we thus use the equal-weighted model average to illustrate the mean response of our ensemble (multimodel mean maps in Figures 1-2), and show the individual responses of each simulation using scatter plots (Figures 3-5). "

**Specific comments:**

**Comment 2** *Page 2, L46 the first mention of CORDEX-AFRICA has no references.*
**Response**: Thank you for this comment, the revised manuscript has been altered accordingly as follows: "... recent CORDEX-AFRICA (Giorgi et al. 2009; Jones et al. 2011; Hewitson et al. 2012; Kim et al. 2014) experiments ...".

**Comment 3**: *Page 8 L 221 There are now published results for West Africa from a convection permitting resolution simulation over Africa. Berthou et al 2019 (Geophy Res Let).*
**Response**: Thank you for this reference, it has been included in the revised manuscript as follows: "For instance, Berthou et al. (2019) have shown that over the West Sahel, future changes in extreme rainfall increase by a factor 5 to 10 at 4.5 km resolution (convection-permitting model allowing a good representation of MCSs), as compared to a factor 2 to 3 at 25 km resolution."

[revised manuscript text omitted]

2012; Kim et al., 2014). All the simulations available online at the time of the analysis have been used.). In this ensemble, 5 Regional Climate Models (RCMs) are used to downscale 10 Global Climate Models (GCMs) under the climate scenario RCP8.5 (Table 1). Out of the 50 combinations possible, only 18 were available, from which 8 use the same RCM. Whereas this imbalance presents the disadvantage to slightly bias the results towards this RCM, it also presents the advantage of representing a large number of GCM, not accessible otherwise. Because the impact of the heterogeneity of the CORDEX-AFRICA GCM-RCM matrix on future precipitation changes is found mostly over Central and West Africa (Dosio et al., 2019), we choose to represent a maximum diversity of RCMs and GCMs. Furthermore, although averaging model output may lead to a loss of signal (such that the true expected change is very likely to be larger than suggested by a model average), there is too little agreement on metrics to separate "good" and "bad" models to objectively weight the models (Knutti et al., 2010). In the following, we thus use the equal-weighted model average to illustrate the mean response of our ensemble (multimodel mean maps in Figures 1-2), and show the individual responses of each simulation using scatter plots (Figures 3-5). The simulations span the period 2006-2099 at a daily mean time step, and cover Africa (24.64°W − 60.28°E; 45.76°S − 42.24°N) at about 50 km (~ 0.5°) spatial resolution in latitude and longitude. For each simulation and each grid cell, daily time series of surface air temperature and precipitation are retrieved.

We evaluate the spatial distribution of daily mean rainfall simulated in the CORDEX models by comparing it to the satellite Climate'Climate Hazards Group Infrared Precipitation with Station data (CHIRPS) dataset CHIRPS)' rainfall dataset (Supplementary Figure 1). The CHIRPS dataset is explicitly designed for monitoring agricultural drought and global environmental change over land. It corresponds to a gridded, quasi-global (50° S− 50° N), high-resolution (0.05°), daily rainfall dataset that covers the time period 1981−2014 (Funk et al., 2015). In addition, we evaluate the statistical distribution of daily mean and yearly mean rainfall simulated in the CORDEX models from in three cities (Ouagadougou Airport (12.35° N, 1.52° W), Dakar-Yoff (14.72° N, 17.51° W, and Accra Kotoka International Airport (5.6° N, 0.17° W), as seen in Figure 1a), by comparing it to the CHIRPS dataset (grid point comparison) and to three near-surface daily rain gauge data extracted from the BADOPLUS dataset, as described in Panthou et al. (2012). Results are shown in Supplementary Figures 1 and 2, and summarized here. Each for the 18 CORDEX simulations 2012) (Supplementary Figure 2). We find that for all the CORDEX simulations, the spatial distribution of mean precipitation and to some extent RR1 satisfactorily reproduces the spatial distribution of the CHIRPS daily mean precipitation, and to some dataset extent RR1 (slightly overestimated in CORDEX, especially along the Guinea Coast ; Supplementary Figure 1(Supplementary Figure 1). The multimodel mean however overestimates the observed RR1, especially along the Guinea Coast. More disagreements are found across models for INT, even though the multimodel mean is in good agreement with the the CHIRPS dataset (Supplementary Figure 1) observation. In DakarIn addition, we

find that in Dakar, the CHIRPS and the BADOPLUS datasets show similar statistical distribution of daily mean and yearly mean precipitation, as well as RR1, daily mean precipitation, yearly mean precipitation, RR1, and to some extent INT (Supplementary Figure 2). In Ouagadougou and Accra however, CHIRPS and the BADOPLUS datasets the two observational datasets show similar statistical distributions for daily mean and yearly mean precipitation, but strongly disagree for RR1 (underestimated in the CHIRPS dataset) and INT (overestimated in the CHIRPS dataset; Supplementary Figure 2). In other words, the CHIRPS dataset shows more frequent but less intense precipitation than the BADOPLUS dataset in Ouagadougou and Accra. Each of the 18 Finally, we find that the CORDEX simulations generally show a similar statistical distribution with than the observations for daily mean and yearly mean precipitation. However but, they tend to overestimate (underestimate) the observed RR1 in Accra and Ouagadougou (Dakar), and underestimate the observed (particularly as compared to the BADOPLUS dataset) INT in all the three locations (Supplementary Figure 2). In other words, the CORDEX simulations show less intense precipitation compared to than the observations (especially the BADOPLUS dataset) in all the three locations, and less (more) frequent precipitation in Dakar (Ouagadougou and Accra) than the observations. Nevertheless, we find that the observations are always included within the range of the 18 CORDEX simulations. Hence, we conclude that the CORDEX simulations compare satisfactorily well with the observations, and can be used for the purpose of our study.

**3. Results**

**3.1 Time evolution**

Trends (2006-2100) of air surface temperature (°C per decade), mean precipitation (mm/day per decade), INT (mm/day per decade), RR1 (days per decade), CDD (days), and CWD (days) are shown in Figure 1, as multimodel mean maps. From Figure 1, 
[revised manuscript text omitted]
. For instance, Berthou et al. (2019) have shown that over the West Sahel, future changes in extreme rainfall increase by a factor 5 to 10 at 4.5 km resolution (convection-permitting model allowing a good representation of MCSs), as compared to a factor 2 to 3 at 25 km resolution. Similarly, the impacts of atmospheric aerosols, particularly abundant over West Africa due to seasonal desert dusts (Konare et al., 2008; N'Datchoh et al., 2018), are only partially accounted for in CORDEX AFRICA due to the simplified parameterization schemes for aerosols in this dataset. However, because aerosols are expected to affect temperature and precipitation in this region (e.g. Konare et al., 2008; N'Datchoh et al., 2018), we suggest that our results are also limited by this simplified representation of aerosols. Additional simulations at higher resolution and using a more complex parameterization scheme for aerosols would be required to identify the impact of MCSs and aerosols on our results, which is beyond the scope of our study.

**Figure Captions.**

[revised manuscript text omitted]

Bichet, A., Diedhiou A., Hingray B., Evin G., N'Datchoh Touré E., Klutse N.A.B., and Kouadio K.: Assessing uncertainties in the regional projections of precipitation in CORDEX-AFRICA (submitted).

Bichet, A., Hingray B., Evin G., Diedhiou A., Kebe C.M.F., and Anquetin S.: Potential impact of climate change on solar resource in Africa for photovoltaic energy: analyses from CORDEX-AFRICA climate experiments, Env. Res. Lett., 14, 12403, 2019.

Berthou, S., Kendon, E., Rowell, D. P., Roberts, M. J., Tucker, S. O., and Stratton, R. A.: Larger future intensification of rainfall in the West African Sahel in a convection-permitting model. Geophysical Research Letters, 46, 13,299–13,307, doi: https://doi.org/10. 1029/2019GL083544, 2019.

Bichet, A., Hingray, B., Evin, G., and Diedhiou, A.: Potential impact of climate change on solar resource in Africa: analyses from CORDEX-AFRICA experiments, submitted.

Collins, M., Knutti, R., Arblaster, J., Dufresne, J.-L., Fichefet, T., Friedlingstein, P., Gao, X., Gutowski, W. J., Johns, T., Krinner, G., Shongwe, M., Tebaldi, C., Weaver, A. J., and Wehner, M.: Long-term Climate Change: Projections, Commitments and Irreversibility. In: Climate Change 2013: The Physical Science Basis. Contribution of Working Group I to the Fifth Assessment Report of the Intergovernmental Panel on Climate Change [Stocker, T. F., Qin, D., Plattner, G.-K., Tignor, M., Allen, S. K., Boschung, J., Nauels, A., Xia, Y., Bex, V., and Midgley, P. M. (eds.)]. Cambridge University Press, Cambridge, United Kingdom and New York, NY, USA, 2013.

Diedhiou, A., Bichet, A., Wartenburger, R., Seneviratne, S. I., Rowell, D. P., Sylla, M. B., Diallo, I., Todzo, S., N'datchoh Touré, E., Camara, M., Ngounou Ngatchah, B., Kane, N. A., Tall, L., and Affholder, F.: Changes in climate extremes over West and Central Africa at 1.5 °C and 2 °C global warming, Env. Res. Lett., 13, 065020., 2018.

Dosio, A., Jones, R.G., Jack, C., Lennard, C., Nikulins, G., and Hewitson, B.: What can we know about future precipitation in Africa? Robustness, significance and added value of projections from a large ensemble of regional climate models, Climate Dynamics, 53, 5833–5858, doi: https://doi.org/10.1007/s00382-019-04900-3, 2019.

Douglas, I., Alam, K., Maghenda, M., McDonnell, Y., Mclean, L., and Campbell, J.: Unjust waters: Climate change, flooding and the urban poor in Africa, Env. and Urban., 20, 187–205, doi:10.1177/0956247808089156, 2008.

Froidurot, S. and Diedhiou, A.: Characteristics of wet and dry spells in the West African monsoon system, Atmos. Sci. Lett., 18, 125−131, 2017.

Funk, C., Peterson, P., Landsfeld, M., Pedreros, D. et al.: The climate hazards infrared precipitation with stations - a new environmental record for monitoring extremes, Sci. Data 2, 150066, 2015.

Giorgi, F., Jones, C., and Asrar, G.: Addressing climate information needs at the regional level. The CORDEX framework, WMO Bulletin, 58, 175-183, 2009.

Giorgi, F., Im, E.-S., Coppola, E., Diffenbaugh, N. S., Gao, X. J., Mariotti, L., and Shi, Y.: Higher hydroclimatic intensity with global warming, J. Climate, 24, 5309–5324, doi:https://doi.org/10.1175/2011JCLI3979.1, 2011.

Giorgi, F., Coppola, E., and Raffaele, F.: A consistent picture of the hydroclimatic response to global warming from multiple indices: Models and observations, J. of Geophys. Res.: Atmosphere, 119, doi:10.1002/2014JD022238, 2014.

Hewitson, B., Lennard, C., Nikulin, G., and Jones, C.: CORDEX-Africa: A unique opportunity for science and capacity building. CLIVAR Exchanges 60, International CLIVAR Project Office, Southampton, UK, pp 6–7, 2012.

Jones, C., Giorgi, F., and Asrar, G.: The coordinated regional downscaling experiment: CORDEX an international downscaling link to CMIP5, CLIVAR Exchanges, 56(16), 34–40, 2011.

Kim, J., Waliser, D.E, Mattmann, C.A., Gooodale, C.E.,Hart, A.F., Zimdars, P.A., Crichon, D.J., Jones,C., Nikulin, G., Hewitson, B., Jack,C., Lennard, C., and Favre, A.: Evaluation of the CORDEX-Africa multi-RCM hindcast: Systematic model errors. Climate Dynamics. 42. 1189-1202. 10.1007/s00382-013-1751-7, 2014.

Klutse, N. A. B., Ajayi, V. O., Gbobaniyi, E. O., Egbebiyi, T. S., Kouadio, K.., Nkrumah, F., Quagraine, K. A., Olusegun, C., Diasso, U., Abiodun, B. J., Lawal, K., Nikulin, G., Lennard, C., and Dosio, A.: Potential impact of 1.5 °C and 2 °C global warming on consecutive dry and wet days over West Africa, Env. Res. Lett., 13, 055013, 2018.

Knutti, R., Furrer, R., Tebaldi, C., Cermak, J., and Meehl, G.A.: Challenges in Combining Projections from Multiple Climate Models. *J. Climate,* 23, 2739–2758, doi: https://doi.org/10.1175/2009JCLI3361.1, 2010.

Konare, A., Zakey, A. S., Solmon, F., Giorgi, F., Rauscher, S., Ibrah, S., and Bi, X.: A regional climate modeling study of the effect of desert dust on the West African monsoon. J Geophys Res 113: D12206, doi:10.1029/2007JD009322, 2008.

Mohan, T. and Rajeevan, M.: Past and future trends of hydroclimatic intensity over the Indian monsoon region, J. Geophys. Res.: Atmospheres, 122, 896–909, 2017.

N'Datchoh Touré, E., Diallo, I., Konaré, A., Silué, S., Ogunjobi, K. O., Diedhiou, A., and Doumbia, M.: Dust induced changes on the West African summer monsoon features. Int J Climatol 38:452–466. https://doi.org/ 10.1002/joc.5187, 2018.

[revised manuscript text omitted]